# A Comparative Study of the Explicit Finite Difference Method and Physics-Informed Neural Networks for Solving the Burgers' Equation

Svetislav Savović [1], Miloš Ivanović [1] and Rui Min [2,*]

1. Faculty of Science, University of Kragujevac, R. Domanovića 12, 34000 Kragujevac, Serbia; savovic@kg.ac.rs (S.S.); mivanovic@kg.ac.rs (M.I.)
2. Center for Cognition and Neuroergonomics, State Key Laboratory of Cognitive Neuroscience and Learning, Beijing Normal University, Zhuhai 519087, China
* Correspondence: rumi@doctor.upv.es

**Abstract:** The Burgers' equation is solved using the explicit finite difference method (EFDM) and physics-informed neural networks (PINN). We compare our numerical results, obtained using the EFDM and PINN for three test problems with various initial conditions and Dirichlet boundary conditions, with the analytical solutions, and, while both approaches yield very good agreement, the EFDM results are more closely aligned with the analytical solutions. Since there is good agreement between all of the numerical findings from the EFDM, PINN, and analytical solutions, both approaches are competitive and deserving of recommendation. The conclusions that are provided are significant for simulating a variety of nonlinear physical phenomena, such as those that occur in flood waves in rivers, chromatography, gas dynamics, and traffic flow. Additionally, the concepts of the solution techniques used in this study may be applied to the development of numerical models for this class of nonlinear partial differential equations by present and future model developers of a wide range of diverse nonlinear physical processes.

**Keywords:** Burgers' equation; finite difference method; physics-informed neural networks

**MSC:** 35K55; 65M06

## 1. Introduction

For many years, both in the fields of fluid mechanics and heat transfer, significant research has been conducted on the analytical techniques and numerical simulations of the non-linear partial differential equations (PDEs) encountered in computational fluid dynamics. The complexity of non-linear PDEs is increasing as more and more real-world characteristics affecting engineering systems are taken into account. The Burgers' equation is one of the most famous equations including both non-linear propagation effects and diffusive effects. Due to its applicability in a variety of domains, such as gas dynamics [1], heat conduction [2], elasticity [3], and solute transport in ground water [4], etc., the study of the general properties of the Burgers' equation has attracted significant interest. In addition, it can be used to evaluate different numerical algorithms. Several studies have been conducted to explore the features of its solution using different analytical and numerical techniques because of its broad range of applications. Using the aid of the Hopf–Cole transformation, Rodin [5] studied a few approximative and exact solutions to the boundary value problem for Burger's equation. With various initial conditions, Benton and Platzman [6] provided 35 different analytic solutions to Burgers' equation. By using group actions on coset bundles, Wolf et al. [7] found a method to extend the analytical solution of Burgers' equation to n-dimensional situations. The solutions were expanded to curvilinear coordinate systems by Nerney et al. [8]. Using Hopf–Cole and

Darboux transformations, Kudryavtsev and Sapozhnikov [9] proposed a method to find the exact solution of the inhomogeneous Burgers' equation. Significant efforts have been made over the past few decades to create reliable numerical techniques for handling the Burgers' equation. The Burgers' equation was transformed into the heat equation by Kutluay et al. [10] using the Hopf–Cole method. Using explicit and exact-EFD solutions, the Burgers' equation transformed into the heat equation with insulated boundary conditions was solved. A two-level, three-point explicit FD scheme that was second-order accurate in time and fourth-order accurate in space was created by Hassanien et al. [11]. The approach is unconditionally stable, according to a von-Neumann stability analysis. In contrast to Bahadir's [12] proposed fully implicit FD scheme, the non-linear system is solved using Newton's method. For solving the Burgers' equation, Kadalbajoo et al. [13] proposed an implicit method. For a numerical simulation of the Burgers' equation, Mukundan and Awastji [14] suggested a numerical approach based on the semi-discretization technique and implicit FDM. The modeled problem must be set up on a mesh (grid) of finite points due to the technique of these numerical approaches. Even though they are thought of as elegant and useful strategies, the more dimensions there are, the more difficult it is to use them. The rise in dimensions is accompanied by an increase in computing processes and resource allocation because of its mesh-based design. As a result of the continued research into artificial intelligence and improvements in computing power, a completely new field of modeling techniques, such as PINN, has been created. Raissi et al. [15] recently demonstrated how PINN can be successfully employed for solving the Burgers' equation. It has been shown that PINN is also an effective tool for solving a nonlinear Schrodinger equation, Allen–Cahn equation, Navier–Stokes equation, and Korteweg–de Vries equation, as well as a high-dimensional inverse problems [15].

In this work, the EFDM and PINN are employed for solving the Burgers' equation. The FDM and PINN are two different approaches for solving PDEs. FDM is a numerical method for approximating a function's derivatives at discrete points inside a domain. The process requires creating a grid of discrete points within the domain, where the function values are calculated. Next, by calculating the difference between the function values at nearby places, the approximate derivatives are derived. The grid step size and approximation order utilized to generate the derivatives have an impact on the FDM's accuracy. PINN is a machine-learning-based approach for solving PDEs. In PINN, a neural network is trained to learn the fundamental physics of a system and approximatively solve a PDE. The residual of the PDE, which measures the discrepancy between the predicted solution and the exact solution, is measured and the neural network is trained to minimize it. The main benefit of PINN is that it can handle complicated boundary conditions and geometries, which can be difficult to model using conventional numerical techniques. On the other hand, it may be computationally expensive and requires a lot of data for training. The selection of hyper-parameters, such as the number of layers and neurons in the neural network, may also have an impact on PINN. In this study, our numerical results for three test problems with various initial conditions and Dirichlet boundary conditions obtained using EFDM and PINN are compared to the analytical solutions reported in the literature.

## 2. The Burgers' Equation

We consider the Burgers' equation:

$$\frac{\partial u(x,t)}{\partial t} = v\frac{\partial u^2(x,t)}{\partial x^2} - u(x,t)\frac{\partial u(x,t)}{\partial x}, \quad x \in [0,1], \ t \in [0,T] \tag{1}$$

with the initial condition:

$$u(x,0) = u_0(x), \ 0 < x < 1 \tag{2}$$

and the boundary conditions:

$$u(0,t) = 0 = u(1,t), \ 0 < t \leq T \tag{3}$$

where $v > 0$ is a parameter, $u\partial u(x,t)/\partial x$ is the non-linear term, and $u_0(x)$ is a given sufficiently smooth function. Burgers' Equation (1) can describe the behavior of fluid flow

and can be used to model various physical phenomena, such as shock waves and turbulence. Then, in Equation (1), $\nu$ is a kinematic viscosity parameter and the term $\partial u(x,t)/\partial t$ represents the time derivative of the velocity, which describes how the velocity of the fluid changes with time. The term $u\partial u(x,t)/\partial x$ represents the non-linear advection of the velocity field, which describes how the fluid carries its own velocity along with it as it flows. The term $\nu\partial^2 u(x,t)/\partial x^2$ represents the diffusion of the velocity field due to the viscosity of the fluid. It describes how the velocity field spreads out over time and space due to the internal friction of the fluid. Therefore, Burgers' equation describes the balance between the advection of the velocity field and the diffusion of the velocity field due to viscosity. When $\nu$ approaches zero, Equation (1) becomes an inviscid Burgers' equation, which is a model for nonlinear wave propagation.

## 3. Explicit Finite Difference Method

Using the EFDM, where the forward FD scheme is used to represent the derivative term $(\partial u(x,t)/\partial t) = (u_i^{j+1} - u_i^j)/\Delta t$ and central FD schemes are used to represent the derivative terms $(\partial u(x,t)/\partial x) = (u_{i+1}^j - u_{i-1}^j)/(2\Delta x)$ and $(\partial^2 u(x,t)/\partial x^2) = (u_{i+1}^j - 2u_i^j + u_{i-1}^j)/(\Delta x)^2$, Equation (1) is written in the following form:

$$\frac{u_i^{j+1} - u_i^j}{\Delta t} = v\frac{u_{i+1}^j - 2u_i^j + u_{i-1}^j}{(\Delta x)^2} - u_i^j\frac{u_{i+1}^j - u_{i-1}^j}{2\Delta x} \tag{4}$$

where $u_i^j \equiv u(x_i, t_j)$ and indexes $i$ and $j$ refer to the discrete step lengths $\Delta x$ and $\Delta t$ for the coordinate $x$ and time $t$, respectively. The grid dimensions in the $x$ and $t$ directions are $K = 1/\Delta x$ and $M = T/\Delta t$, respectively. Using the FD scheme, the initial condition (2) and boundary conditions (3) are given as:

$$u_i^0 = u_0(x_i); \ 0 < x_i < 1, \ i = 1, 2, \ldots, K \ (t = 0) \tag{5}$$

$$u_0^j = 0 = u_K^j, \ j = 0, 1, \ldots, M \ (x = 0 \text{ and } x = 1) \tag{6}$$

Equation (4) represents a formula for $u_i^{j+1}$ at the $(i, j+1)$th mesh point in terms of the known values along the $j$th time row. The truncation error for the difference Equation (4) is $O(\Delta t, (\Delta x)^2)$. The truncation error can be decreased using small enough values of $\Delta t$ and $\Delta x$ until the accuracy attained is within the error tolerance.

## 4. Physics-Informed Neural Networks

*4.1. The Basic Concept of Physics-Informed Neural Networks in Solving PDEs*

A machine learning method called the PINN can be used to approximatively solve PDEs. A general form of PDEs with corresponding initial and boundary conditions is:

$$\begin{aligned} &\frac{\partial u(x,t)}{\partial t} + N[u(x,t)] = 0, \ x \in \Omega, \ t \in [0, T] \\ &u(x, t = 0) = h(x), \ x \in \Omega \\ &u(x,t) = g(x,t), \ x \in \Omega_g, \ t \in [0, T] \end{aligned} \tag{7}$$

Here, $N$ is a differential operator, $x \in \Omega \subseteq R^d$ and $t \in R$ represent spatial and temporal dimensions, respectively, $\Omega \subseteq R^d$ is a computational domain, $\Omega_g \subseteq \Omega$ is a computational domain of the exposed boundary conditions, and $u(x,t)$ is the solution of the PDEs with the initial condition $h(x)$ and boundary conditions $g(x,t)$.

In the original formulation [16], an approximator network and a residual network are the two subnets that make up PINN. After receiving the input $(x,t)$ and going through the training process, the approximator network outputs an approximate solution $\widehat{u}(x,t)$. A grid of points, referred to as collocation points, sampled at random or on a regular basis from the simulation domain, is used by the approximator network to train. The weights and biases of the approximator network make up a set of trainable parameters, trained by minimizing a composite loss function of the following form:

$$L = L_r + L_0 + L_b \tag{8}$$

where:

$$
\begin{aligned}
L_r &= \frac{1}{N_r} \sum_{i=1}^{N_r} \left| u(x^i, t^i) + N[u(x^i, t^i)] \right|^2 \\
L_0 &= \frac{1}{N_0} \sum_{i=1}^{N_0} \left| u(x^i, t^i) - h^i)] \right|^2 \\
L_b &= \frac{1}{Nb} \sum_{i=1}^{N_b} \left| u(x^i, t^i) - g^i)] \right|^2
\end{aligned}
\tag{9}
$$

Here, $L_r$, $L_0$, and $L_b$ represent residuals of the governing equations, initial, and boundary conditions, respectively. $N_r$, $N_0$, and $N_b$ are the numbers of the mentioned collocation points of the computational domain, initial, and boundary conditions, respectively. The residual network, a non-trainable component of the PINN model, calculates these residuals. PINN needs derivatives of the outputs with respect to the inputs $x$ and $t$ to calculate the residual $L_r$. Such a calculation is performed through automated differentiation, which relies on the fact that combining derivatives of the constituent operations by the chain rule produces the derivative of the entire composition. This technique is a key enabler for the development of PINNs and is the main element that differentiates PINNs from comparable efforts in the early 1990's, which relied on the manual derivation of back-propagation rules. Nowadays, automatic differentiation capabilities are well-implemented in most deep learning frameworks, such as TensorFlow and PyTorch, avoiding tedious derivations or numerical discretization while computing derivatives of all orders in space–time.

A schematic of the PINN is demonstrated in Figure 1, in which a simple partial differential equation $\partial f / \partial x + \partial f / \partial y = u$ is used as an example. The approximator network is used to approximate the solution $u(x, t)$ which then goes to the residual network to calculate the residual loss $L_r$, boundary condition loss $L_b$, and initial condition loss $L_0$. The weights and biases of the approximator network are trained using a composite loss function consisting of the residuals $L_r$, $L_0$, and $L_b$ through a gradient descent technique based on the back-propagation.

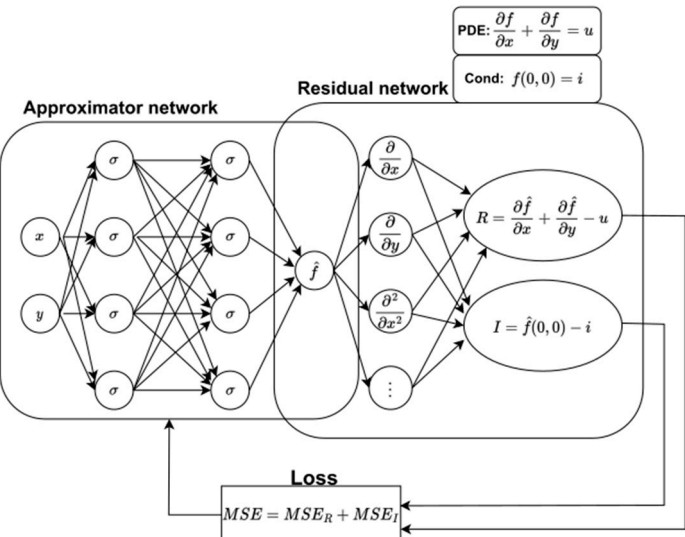

**Figure 1.** The architecture of a PINN and the standard training loop of a PINN constructed for solving a simple partial differential equation, where PDE and Cond denote governing equations, while *R* and *I* represent their residuals. The approximator network is subjected to a training process and provides an approximate solution. The residual network is a non-trainable part of PINN capable of computing derivatives of the approximator network outputs with respect to the inputs, resulting in the composite loss function, denoted by MSE.

### 4.2. Implementation of PINN in Solving the Burgers' Equation

To conduct the PINN model development for solving the Burgers' equation, we employed the DeepXDE library [17]. Our PINN has two inputs $(x, t)$ and contains three layers consisting of 20 neurons each. All neurons exhibit tanh activation. The set of collocation points consists of three subsets. The largest subset of 5080 contains the collocation points that belong to a general problem domain. The second and third subsets are smaller, with 320 and 160 collocation points, and their purpose is to enforce the boundary and initial conditions, respectfully. These conditions are identical in all our test problems. The PINN training process consists of two phases. In the first phase, we optimize the weights and biases using the Adam algorithm for 15,000 epochs with a learning rate of $10^{-3}$. In the second phase, after a "global" search is completed, the Limited Memory Broyden–Fletcher–Goldfarb–Shanno algorithm (L-BFGS) acts to get closer to the optimal solution according to [18]. The whole training process takes approximately 50 s on an nVidia Tesla T4 GPU accelerator. Practically speaking, it is very likely that using various hyper-parameters, such as various activation functions, training techniques, and varying PINN topologies, will result in better solutions. However, since finding hyper-parameters is a tedious and time-consuming process and is outside the scope of our study, we selected the hyper-parameter values that were most prevalent in the Burgers' problem literature.

The purpose of this work is to compare the accuracy of the numerical results obtained using the EFDM and PINN for three test problems of Burgers' equation with respect to the analytical solutions available in the literature.

## 5. Results and Discussion

To illustrate the accuracy of the EFD scheme and PINN, several numerical computations are carried out for three test problems.

**Test problem 1:** Consider the Burgers' equation:

$$\frac{\partial u(x,t)}{\partial t} = v\frac{\partial u^2(x,t)}{\partial x^2} - u(x,t)\frac{\partial u(x,t)}{\partial x}, \quad x \in [0,1], \ t \in [0,T] \tag{10}$$

with the initial condition:

$$u(x,0) = \sin(\pi x), \ 0 < x < 1 \tag{11}$$

and the boundary conditions:

$$u(0,t) = 0 = u(1,t), \ 0 < t \le T \tag{12}$$

The analytical solution of the problem is given as [19]:

$$u(x,t) = 2\pi v\frac{\sum\limits_{n=1}^{\infty} C_n \exp(-n^2\pi^2 vt)n\sin(n\pi x)}{C_0 + \sum\limits_{n=1}^{\infty} C_n \exp(-n^2\pi^2 vt)\cos(n\pi x)} \tag{13}$$

where:

$$C_0 = \int\limits_0^1 \exp\left\{-\frac{1}{2\pi v}[1 - \cos(\pi x)]\right\}dx \tag{14}$$

$$C_n = 2\int\limits_0^1 \exp\left\{-\frac{1}{2\pi v}[1 - \cos(\pi x)]\right\}\cos(n\pi x)dx \tag{15}$$

Equation (4) represents the EFD scheme of this test problem, the boundary conditions are given in (6), and the initial condition (11) becomes:

$$u_i^0 = \sin(\pi x_i); \ 0 < x_i < 1, \ i = 1, 2, \ldots, K \ (t = 0) \tag{16}$$

Figures 2 and 3 compare our numerical solutions of the Burgers' Equation (10) obtained using the EFD scheme (step lengths are $\Delta x = 0.01$ and $\Delta t = 0.0001$) and PINN, with analytical solutions (13) at different times T, for a kinematic viscosity parameter $\nu = 0.5$ and 0.05, respectively. A good agreement between our numerical solutions and analytical solutions can be seen. Because Figures 2 and 3 are insufficient for an exact comparison of the two numerical methods, we considered the root mean square error defined by:

$$Error = \sqrt{\frac{1}{N}\sum_{i=1}^{K}\left(u_i^{method} - u_i^{analit}\right)^2} \tag{17}$$

where $K$ is the total number of observed points along the $x$ axis. Equation (17) was taken as the error function, representing an accuracy evaluation of the method. As the error value decreases, the method gives a better distribution $u(x,t)$ over a given time interval. As an additional illustration, Figure 4 shows the physical behavior of the EFD and PINN solutions of Test problem 1 in 3D at different times for $\nu = 0.05$.

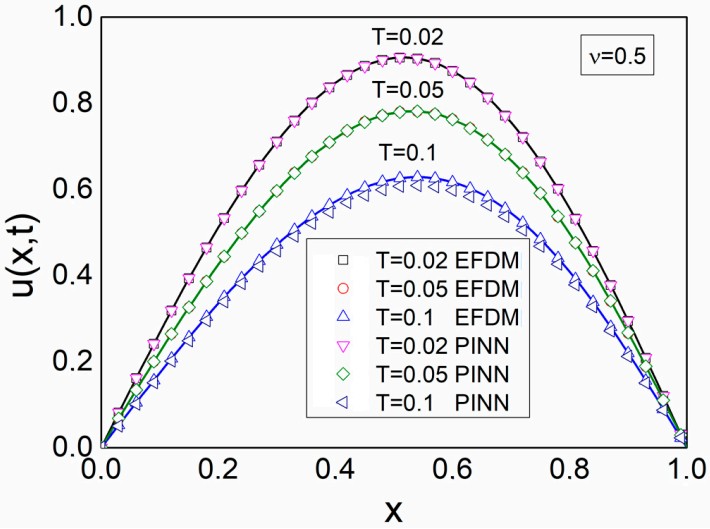

**Figure 2.** EFD and PINN solutions (open symbols) compared to analytical solutions (solid lines) of Test problem 1 at different times $T = 0.02$, 0.05, and 0.1 for $\nu = 0.5$.

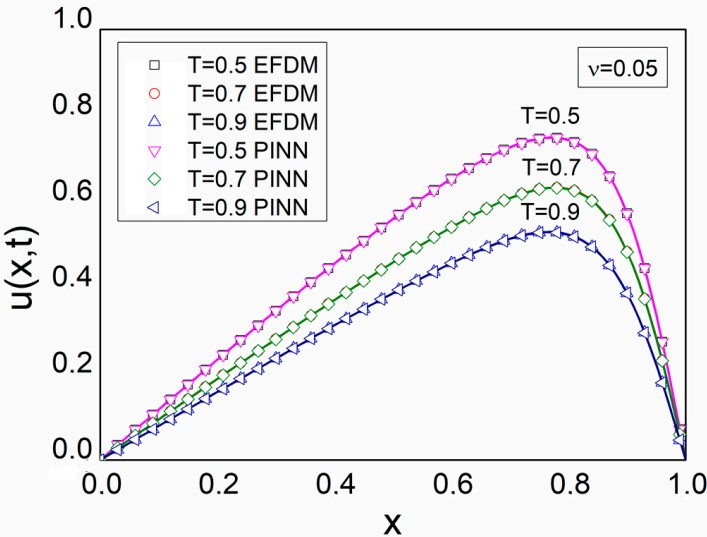

**Figure 3.** EFD and PINN solutions (open symbols) compared to analytical solutions (solid lines) of Test problem 1 at different times $T = 0.5$, 0.7, and 0.9 for $\nu = 0.05$.

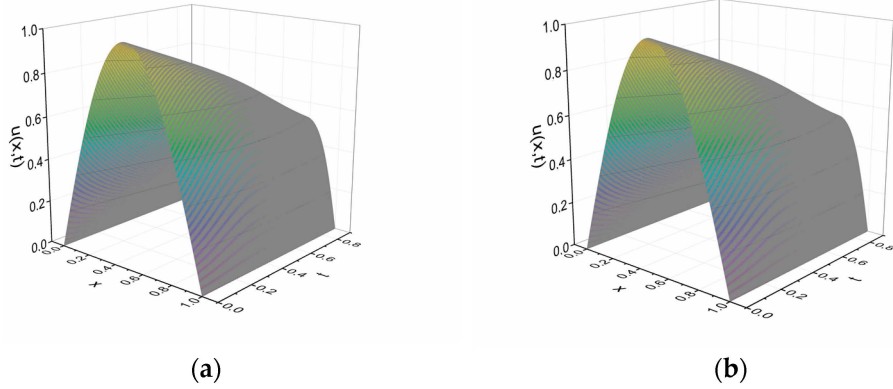

**Figure 4.** (**a**) EFD and (**b**) PINN solutions of Test problem 1 in 3D at different times for $v = 0.05$.

Table 1 represents the accuracy of the EFDM and PINN for two kinematic viscosity parameters $v$. It can be noted that the EFDM provides a better match with the analytical solution.

**Table 1.** The accuracy of EFDM and PINN for different kinematic viscosity coefficients $v$.

|  | $T$ | Error (EFDM) | Error (PINN) |
| --- | --- | --- | --- |
| | 0.02 | $5.14 \times 10^{-7}$ | $2.56 \times 10^{-5}$ |
| $v = 0.5$ | 0.05 | $5.07 \times 10^{-7}$ | $4.96 \times 10^{-5}$ |
| | 0.1 | $5.43 \times 10^{-5}$ | $9.51 \times 10^{-5}$ |
| | 0.5 | $4.43 \times 10^{-7}$ | $7.09 \times 10^{-6}$ |
| $v = 0.05$ | 0.7 | $2.38 \times 10^{-7}$ | $1.46 \times 10^{-6}$ |
| | 0.9 | $7.03 \times 10^{-8}$ | $1.02 \times 10^{-6}$ |

**Test problem 2:** Consider the Burgers' equation:

$$\frac{\partial u(x,t)}{\partial t} = v\frac{\partial u^2(x,t)}{\partial x^2} - u(x,t)\frac{\partial u(x,t)}{\partial x}, \quad x \in [0,1], \ t \in [0,T] \tag{18}$$

with the initial condition:

$$u(x,0) = 4x(1-x), \ 0 < x < 1 \tag{19}$$

and the boundary conditions:

$$u(0,t) = 0 = u(1,t), \ 0 < t \leq T \tag{20}$$

The analytical solution of the problem is given as [19]:

$$u(x,t) = 2\pi v\frac{\sum\limits_{n=1}^{\infty} D_n \exp(-n^2\pi^2 vt) n \sin(n\pi x)}{D_0 + \sum\limits_{n=1}^{\infty} D_n \exp(-n^2\pi^2 vt) \cos(n\pi x)} \tag{21}$$

where:

$$D_0 = \int_0^1 \exp\left\{-\frac{1}{3v}[x^2(3-2x)]\right\} dx \tag{22}$$

$$D_n = 2\int_0^1 \exp\left\{-\frac{1}{3v}[x^2(3-2x)]\right\} \cos(n\pi x) dx \tag{23}$$

Equation (4) represents the EFD solution of this test problem, the boundary conditions are given in Equation (6), and the initial condition (19) becomes:

$$u_i^0 = 4x_i(1 - x_i); \ 0 < x_i < 1, \ i = 1, 2, \ldots, K \ (t = 0) \tag{24}$$

Figures 5 and 6 compare our numerical solutions of the Burgers' Equation (18) obtained using the EFD scheme (step lengths are $\Delta x = 0.01$ and $\Delta t = 0.0001$) and PINN with analytical solutions (21) at different times T for a kinematic viscosity parameter $\nu = 0.5$ and 0.1. A good agreement between these solutions can be seen. Figure 7 depicts the physical behavior of the EFD and PINN solutions of Test problem 2 in 3D at different times for $\nu = 0.5$.

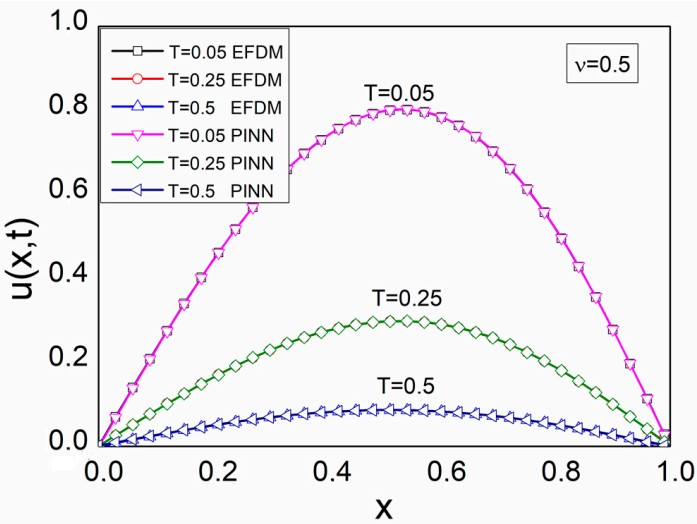

**Figure 5.** EFD and PINN solutions (open symbols) compared to analytical solutions (solid lines) of Test problem 2 at different times $T = 0.05$, 0.25, and 0.5 for $\nu = 0.5$.

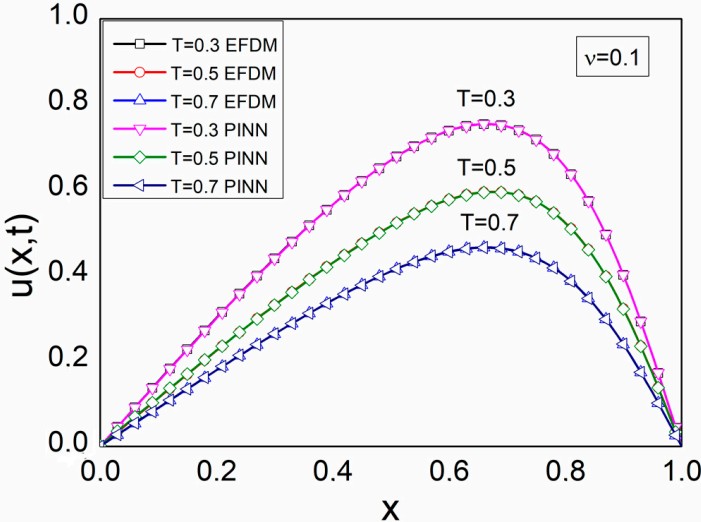

**Figure 6.** EFD and PINN solutions (open symbols) compared to analytical solutions (solid lines) of Test problem 2 at different times $T = 0.3$, 0.5, and 0.7 for $\nu = 0.1$.

The accuracy of the EFDM and PINN for two kinematic viscosity parameters $v$ is given in Table 2. It can be seen that the numerical results obtained using the EFDM are in better agreement with the analytical solution.

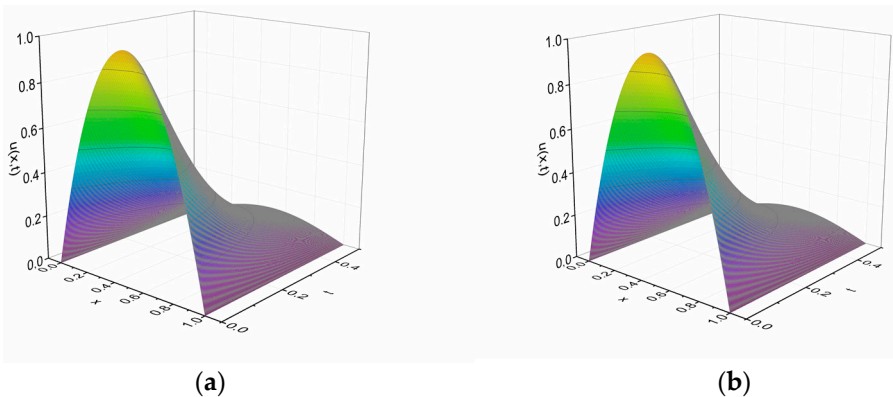

**Figure 7.** (**a**) EFD and (**b**) PINN solutions of Test problem 2 in 3D at different times for $v = 0.5$.

**Table 2.** The accuracy of EFDM and PINN for different kinematic viscosity coefficients $v$.

|  | $T$ | Error (EFDM) | Error (PINN) |
|---|---|---|---|
|  | 0.05 | $5.36 \times 10^{-8}$ | $2.16 \times 10^{-4}$ |
| $v = 0.5$ | 0.25 | $2.37 \times 10^{-7}$ | $2.27 \times 10^{-6}$ |
|  | 0.5 | $1.14 \times 10^{-7}$ | $1.57 \times 10^{-4}$ |
|  | 0.3 | $3.80 \times 10^{-9}$ | $9.09 \times 10^{-7}$ |
| $v = 0.1$ | 0.5 | $6.19 \times 10^{-7}$ | $1.65 \times 10^{-4}$ |
|  | 0.7 | $4.34 \times 10^{-7}$ | $4.79 \times 10^{-5}$ |

**Test problem 3:** Consider the Burgers' equation:

$$\frac{\partial u(x,t)}{\partial t} = v \frac{\partial u^2(x,t)}{\partial x^2} - u(x,t) \frac{\partial u(x,t)}{\partial x}, \ x \in [0,1], \ t \in [0,T] \tag{25}$$

with the initial condition:

$$u(x,0) = \frac{2v\pi \sin(\pi x)}{m + \cos(\pi x)}, \ 0 < x < 1 \tag{26}$$

where $m > 1$ is a parameter, and the boundary conditions:

$$u(0,t) = 0 = u(1,t), \ 0 < t \leq T \tag{27}$$

The analytical solution of the problem is given as [20]:

$$u(x,t) = \frac{2v\pi \exp(-\pi^2 vt) \sin(\pi x)}{m + \exp(-\pi^2 vt) \cos(\pi x)}, \ m > 1 \tag{28}$$

Equation (4) represents the EFD solution of this test problem, the boundary conditions are given in Equation (6), and the initial condition (26) becomes:

$$u_i^0 = \frac{2v\pi \sin(\pi x_i)}{m + \cos(\pi x_i)}, \ 0 < x_i < 1, \ i = 1, 2, \dots, K \ (t = 0) \tag{29}$$

Figures 8 and 9 compare our numerical solutions of the Burgers' Equation (25) obtained using the EFD scheme (step lengths are $\Delta x = 0.01$ and $\Delta t = 0.0001$) and PINN, with analytical solutions (28) (we used parameter $m = 2$) at different times $T$ for a kinematic viscosity parameter $v = 0.5$ and 0.02. A good agreement between these solutions can be seen. One can observe from Figure 10a the physical behavior of the EFD and PINN solutions of Test problem 1 in 3D at different times for $v = 0.02$.

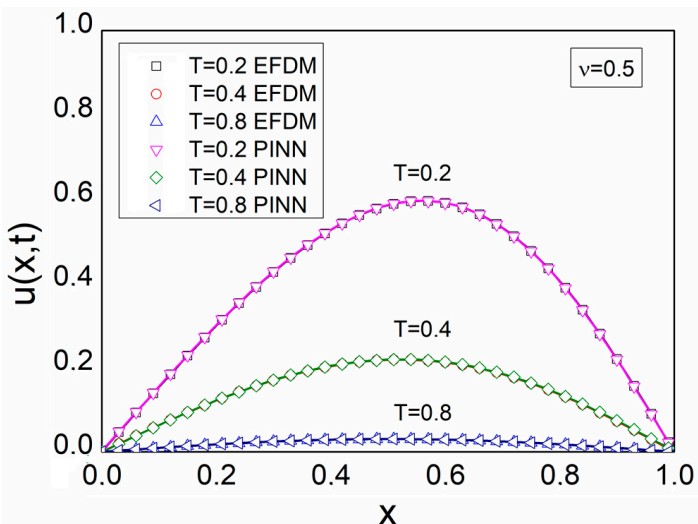

**Figure 8.** EFD and PINN solutions (open symbols) compared to analytical solutions (solid lines) of Test problem 3 at different times $T$ = 0.2, 0.4, and 0.8 for $\nu$ = 0.5.

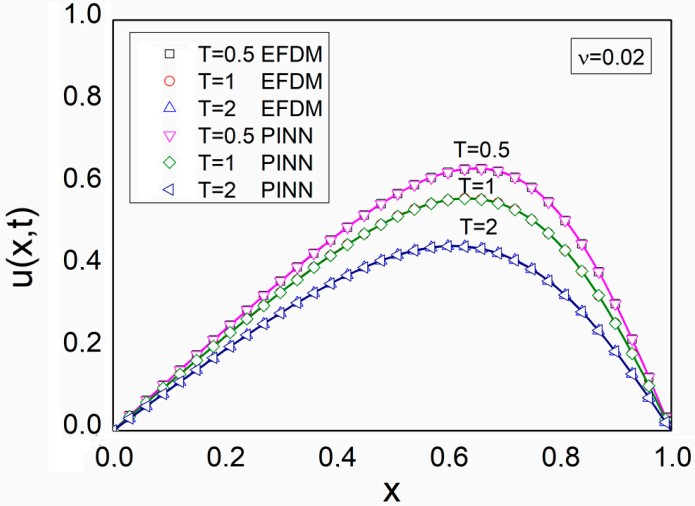

**Figure 9.** EFD and PINN solutions (open symbols) compared to analytical solutions (solid lines) of Test problem 3 at different times $T$ = 0.5, 1, and 2 for $\nu$ = 0.02.

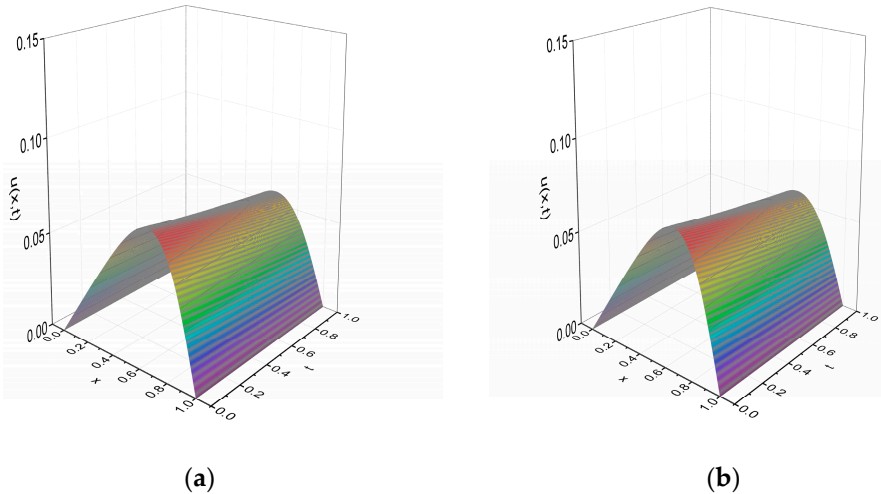

**Figure 10.** (**a**) EFD and (**b**) PINN solutions of Test problem 3 in 3D at different times for $\nu$ = 0.02.

Table 3 represents the accuracy of the EFDM and PINN for two kinematic viscosity parameters $\nu$. It can be noted that the EFDM provides a better match with the analytical solution. It can also be seen that, with a decreasing kinematic viscosity parameter, the error decreases both for the EFDM and PINN.

**Table 3.** The accuracy of EFDM and PINN for different kinematic viscosity coefficients $\nu$.

|  | $T$ | Error (EFDM) | Error (PINN) |
|---|---|---|---|
| | 0.2 | $6.05 \times 10^{-5}$ | $9.72 \times 10^{-4}$ |
| $\nu = 0.5$ | 0.4 | $6.07 \times 10^{-5}$ | $7.56 \times 10^{-4}$ |
| | 0.8 | $1.24 \times 10^{-5}$ | $2.32 \times 10^{-4}$ |
| | 0.5 | $3.85 \times 10^{-6}$ | $2.15 \times 10^{-5}$ |
| $\nu = 0.02$ | 1 | $7.45 \times 10^{-6}$ | $2.33 \times 10^{-5}$ |
| | 2 | $1.12 \times 10^{-5}$ | $3.27 \times 10^{-4}$ |

It is worth noting that, in this work, we used a small enough value of $\Delta t$ in order to achieve the stability of the EFD scheme. As an illustration, a similar situation arises for solving a Lax–Wendroff-modified differential equation for linear and nonlinear advection [21]. Alternatively, one can adopt unconditionally stable algorithms, such as the unconditionally positive finite difference method [22], Dufort–Frankel [23], and Leapfrog–Hopscotch scheme [24]. On the other hand, one should also mention that the Burgers' equation with Neumann boundary conditions was solved using a domain decomposition method [25].

## 6. Conclusions

In solving nonlinear parabolic differential equations of the Burgers' type, we compared our numerical results obtained using EFDM and PINN with the analytical solutions reported in the literature. To the best of our knowledge, for the first time, we compared, in this work, the accuracy of the EFDM and PINN for solving the Burgers' equation with three different initial conditions and Dirichlet boundary conditions. We demonstrated that, although both approaches yield a very good agreement with analytical solutions, the EFD scheme with sufficiently fine step lengths $\Delta x$ and $\Delta t$ showed a higher accuracy compared to the numerical solutions calculated using PINN. Since all the numerical results obtained by the above methods showed a reasonably good agreement with the analytical solutions, both methods can therefore be competitive and worth recommendation. Current and future developers of models for a broad range of various nonlinear physical processes may draw on the ideas of the solution methods employed in this study to further develop numerical models for nonlinear partial differential equations. The presented results are important when modeling various nonlinear physical processes using the Burgers' equation, including those which arise in gas dynamics, traffic flow, chromatography, and flood waves in rivers.

**Author Contributions:** Conceptualization, S.S. and M.I.; methodology, S.S.; software, S.S. and M.I.; validation, S.S., M.I. and R.M.; formal analysis, R.M.; investigation, M.I.; resources, R.M.; data curation, M.I.; writing—original draft preparation, S.S. and M.I.; writing—review and editing, S.S. and M.I.; visualization, R.M.; supervision, S.S.; project administration, R.M.; funding acquisition, R.M. All authors have read and agreed to the published version of the manuscript.

**Funding:** This research was funded by the Serbian Ministry of Science, Technological Development and Innovations (Agreement No. 451-03-47/2023-01/200122) and by grant from Science Fund of the Republic of Serbia (Agreement No. CTPCF-6379382). the National Natural Science Foundation of China (62111530238, 62003046); Guangdong Basic and Applied Basic Research Foundation (2021A1515011997); Special project in key field of Guangdong Provincial Department of Education (2021ZDZX1050); The Innovation Team Project of Guangdong Provincial Department of Education (2021KCXTD014).

**Data Availability Statement:** Data are contained within the article.

**Conflicts of Interest:** The authors declare no conflict of interest.

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
