# Peer review of "A Comparative Study of the Explicit Finite Difference Method and Physics-Informed Neural Networks for Solving the Burgers’ Equation"

_axioms, doi:10.3390/axioms12100982_

Round 1

Reviewer 1 Report

The paper must be improved

Reviewer 2 Report

This paper is very interesting and mostly very well-written, definitely worth to be published. With my comments, I would like to help to improve it further. My specific comments are as follows:

1. In line 46, it is not completely clear that the sentence starting as "Using explicit and exact-EFDM, the transformed" belongs to the previous (probably) or the following sentence. In the former case, the authors should mention whether the "two-level, three-point FD scheme" in the following sentence is explicit or implicit.

2. Line 76: Do the authors mean exact or true solution by "actual" solution?

3. In Eq. (1) the authors use x_max as the upper boundary of the space interval. Two lines later, however, this boundary is taken as 1 without specifying the value of x_max. Either use x_max or 1 everywhere. The same problem arises later as well.

4. line 91. "time derivative of the velocity,". Do u always have the meaning velocity in all the applications? If not, then the authors should state before that sentence that they will call u as velocity for the sake of simplicity or something like this.

5. Line 103: The 2Deltax should be in brackets in the denominator to avoid confusion.

6. Either i should start from 0 instead of 1 in (5) or j should start from 0, since currently, u_0^0 is not given.

7. In the first equation of (9), why there is "u(xi,t1)+" instead of the time derivative of u followed by a minus?

8. Line 130. The last N should be N_b, with b as boundary.

9. Line 166. 10-3 should perhaps be 10^(-3).

10. Line 186: "Equation (4) represents the EFD solution of this test problem". No, it is only the scheme to obtain the solution.

11. Line 221: Condition (18) should be (19).

12. The paragraph before the Conclusion section (from line 277) should be merged into the Conclusion section.

12+1. Finally, but most importantly: The study suggests that the simplest explicit (Euler) method can be recommended to solve this and similar problems. There is no word about its main disadvantage, conditional stability. The authors "accidentally" used time step sizes below the stability threshold (CFL limit or mesh Fourier number) without mentioning that above this, the solution is expected to blow up. I think the authors should write about this problem in the introduction, at least mentioning some of the algorithms which are unconditionally stable for the diffusion equation (UPFD, Dufort-Frankel, odd-even hopscotch, leapfrog-hopscotch, etc.) and (in the future) should be adapted for Burgers' equation as well.

I found very few mistakes, such as:

Line 34: "researches... to exploring" explore.

Line 61: Instead of "like PINN", I would write "such as PINN"

Round 2

Reviewer 1 Report

No comments

Reviewer 2 Report

The paper is sufficiently improved and can be published in its current form.